# Illumination Temporal Fluctuation Suppression for Single-Pixel Imaging

**DOI:** 10.3390/s23031478

**Published:** 2023-01-28

**Authors:** Han Wang, Mingjie Sun, Lailiang Song

**Affiliations:** School of Instrumentation Science and Optoelectronic Engineering, Beihang University, Beijing 100191, China

**Keywords:** differential, Hadamard matrix, illumination fluctuation, normalization, single-pixel imaging

## Abstract

Single-pixel cameras offer improved performance in non-visible imaging compared with modern digital cameras which capture images with an array of detector pixels. However, the quality of the images reconstructed by single-pixel imaging technology fails to match traditional cameras. Since it requires a sequence of measurements to retrieve a single image, the temporal fluctuation of illumination intensity during the measuring will cause inconsistence for consecutive measurements and thus noise in reconstructed images. In this paper, a normalization protocol utilizing the differential measurements in single-pixel imaging is proposed to reduce such inconsistence with no additional hardware required. Numerical and practical experiments are performed to investigate the influences of temporal fluctuation of different degrees on image quality and to demonstrate the feasibility of the proposed normalization protocol. Experimental results show that our normalization protocol can match the performance of the system with the reference arm. The proposed normalization protocol is straightforward with the potential to be easily applied in any temporal-sequence imaging strategy.

## 1. Introduction

As the pursuit of better imaging quality never ends, the rapid growth in the number of pixels in digital cameras is not surprising. In the meantime, the technology using just a single-pixel detector is still an important part of recent research [1,2,3,4,5,6,7,8,9,10,11,12,13,14,15,16,17,18,19,20,21,22,23,24,25,26,27,28,29,30,31]. This primitive technology reignited a lot of interest when the work of Todd Pittman in 1995 [1], also known as ghost imaging, demonstrated that an optical imaging system using two bucket detectors without spatial resolution can reconstruct an image of an object, and even suggested that such imaging scheme exploited the quantum entanglement of the bi-photon source. Later, the substantial equivalence between the classical and quantum approaches of ghost imaging was demonstrated [2,3,4,5,6,7,8,9,10]. In the past decades, computational ghost imaging has been the major approach for ghost imaging researches [11,12,13,14], and different imaging methods such as correlated imaging and ghost imaging, which are essentially the same, are collectively referred to as single-pixel imaging.

In a single-pixel imaging system, a spatial light modulator (SLM) is applied with predesigned time-varying patterns. The light field structured by SLM interacts with the object and is recorded by a non-spatially resolving detector. Normally with a sequence of measurements, the image can be reconstructed by these patterns and corresponding observed intensities. It seems not so convenient to capture an image by single-pixel imaging technology. However, it has been demonstrated that single-pixel imaging has the advantage over conventional imaging on detection efficiency, sensitivity at non-visible wavelengths, and timing resolution. Before a new cutting-edge detection technology can be manufactured into a cost-effective array, single-pixel imaging is the most convenient, if not the only, method to perform imaging with the so-called cutting-edge detector. Therefore, single-pixel imaging systems can be a simple and cost-effective choice in non-conventional imaging regimes where conventional cameras are considerably complicated and expensive, including but not limited to infrared imaging [15,16,17], X-ray imaging [18,19,20], terahertz imaging [21,22,23], multispectral imaging [24,25,26], and hyperspectral imaging [27,28,29].

Despite all these advantages, there is still a long way to go before single-pixel imaging can step into the practical stage. The signal-to-noise ratio (SNR) of the reconstructed image has been a major limitation against its development. A few studies aiming at improving the SNR of single-pixel imaging technology have been developed in recent years [30,31,32,33], while the quality of images reconstructed by single-pixel imaging is still not satisfying. Rather than acquiring all pixel information at once like the conventional pixelated-detector-based imaging strategy, single-pixel imaging requires multiple measurements, thus it is also disturbed by inconsistence caused by illumination fluctuations among measurements besides detector noise. In addition, the illumination noise tends to dominate at high illuminance level, hence it is a prior problem to be solved in most scenarios. Endeavors have been made to address the spatial non-uniformity of illumination [34,35,36]. However, temporal fluctuation influence, which can be partially suppressed by the widely used differential measurement [30,31], has not been intensively investigated.

In this work, we analyze the temporal fluctuation noise in Differential Hadamard Single-pixel Imaging (DHSI) and introduce a normalization protocol to fully suppress the noise brought by the fluctuation of illumination intensity. The sum of signals corresponding to two complementary patterns is used as a weighting factor to normalize the corresponding signals before reconstructing images. The improvements in image quality of numerical and experimental results indicate the feasibility of the proposed protocol. This work offers more potential of DHSI without any hardware addition and gains significant improvement of SNR at the price of a little more calculation.

## 2. Materials and Methods

### 2.1. Principles of DHSI

Considering a pixelated 2D image as a matrix consisting of N unknown variables, single-pixel imaging technology gives a certain way to build a system of linear equations which can solve these N elements. The number of the equations is the number of the patterns M. The 2D image can be transformed into a column vector $O={\left[O_{1},O_{2},\ldots ,O_{N}\right]}^{T}$. Moreover, the i’th pattern displayed can be represented as a row vector $P_{i}=\left[P_{i,1},P_{i,2},\ldots ,P_{i,NN}\right]$, where i=1,2,…,M. Each independent measurement represents one of the equations, and the measured intensity of correlation between the object O and the pattern $P_{i}$ can be formulated as
(1)$$S_{i}=P_{i}O$$
where subscript i is used to denote the pattern sequence number. After all M measurements are performed, the linear equation set is formed as
(2)S=PO
where $S={\left[S_{1},S_{2},\ldots ,S_{M}\right]}^{T}$ and $P={\left[P_{1}^{T},P_{2}^{T},\ldots ,P_{M}^{T}\right]}^{T}$. The reconstruction of the image is to solve this system of linear equations, and the solution is determined by the choice of patterns. There are various choices of patterns so far, among which the Hadamard basis is popular for its orthogonality and binary nature [37,38]. The typical Hadamard matrix is constructed of order 2k for every non-negative integer k, namely
(3)$$\begin{array}{c} H_{2}=\left[\begin{array}{cc} 1 & 1\\ 1 & -1 \end{array}\right]\\ \begin{array}{c} \vdots \\ H_{2^{k}}=\left[\begin{array}{cc} H_{2^{k-1}} & H_{2^{k-1}}\\ H_{2^{k-1}} & -H_{2^{k-1}} \end{array}\right]=H_{2}⊗H_{2^{k-1}} \end{array} \end{array}$$

In Hadamard-based single-pixel imaging M=N and $HH^{T}=NE_{N}$ such that image reconstruction can be performed without matrix inversion by
(4)$$O=P^{-1}S=H^{-1}S=\frac{1}{N}H^{T}S$$

The original Hadamard matrix contains negative values which cannot be displayed by SLM. Therefore, Hadamard Single-pixel Imaging (HSI) is commonly performed in a manner of differential measurements. As illustrated in Figure 1, one Hadamard pattern is divided into two patterns that are inverse to each other. These two groups of patterns are usually generated by
(5)$$\begin{array}{c} P_{i+,j}=\frac{1}{2}\left[1+H_{i,j}\right]\\ P_{i-,j}=\frac{1}{2}\left[1-H_{i,j}\right] \end{array}$$
where $H_{i,j}$ represents the element at the corresponding position of the Hadamard matrix. The data used to reconstruct the image are the differential between the two groups of measurements
(6)$$S=PO=\left(P_{+}-P_{-}\right)O=S_{+}-S_{-}$$

### 2.2. Noise in DHSI

In a single-pixel imaging system, noise can be classified into two categories: multiplicative noise and additive noise. Suppose the illumination intensity is I and the additive noise is n, the intensity measured by the single-pixel detector should be
(7)$$S_{i}^{*}=I_{i}P_{i}O+n_{i}=I_{i}S_{i}+n_{i}$$
where $S_{i}^{*}$ is the observed value and $S_{i}$ is the true value. In differential measurement
(8)$$S_{i}^{*}=S_{i+}^{*}-S_{i-}^{*}=I_{i+}S_{i+}^{*}-I_{i-}S_{i-}^{*}$$

Differential measurement is conducive to the suppression of additive noise. However, it is not helpful in the suppression of multiplicative noise. The multiplicative noise is mainly caused by the fluctuations in the ambient illumination level between different mask pattern displays, and the nature of the temporal fluctuations is the light source’s output power fluctuations, which may be caused by fluctuations of temperature, drive current, and other factors. According to Equation (4), the reconstructed image turns out to be
(9)$$O^{*}=P^{-1}IPO$$
where $I=\left[\begin{array}{cccc} \frac{I_{1+}+I_{1-}}{2} & 0 & \cdots & 0\\ 0 & \frac{I_{2+}+I_{2-}}{2} & \cdots & 0\\ \vdots & \;\vdots & \ddots & \vdots \\ 0 & 0 & 0 & \frac{I_{N+}+I_{N-}}{2} \end{array}\right]$.

The root mean squared error (***RMSE***) of the reconstructed image can be formulated as
(10)$$RMSE=\frac{1}{\sqrt{N}}{∥O^{*}-O∥}_{2}=\frac{1}{\sqrt{N}}{∥\left(P^{-1}IP-E\right)O∥}_{2}$$

As seen in Equation (10), the noise of the reconstructed image depends on the consistency of illumination intensity corresponding to the measurement.

### 2.3. Normalization Protocol

Temporal fluctuations of illumination intensity cause inconsistence in measurements. As shown in Figure 2a,b, fluctuations of the illumination result in different levels of deviation from the ideal value in different measurements. Temporal calibration is required to reduce such inconsistence. Existing temporal correction methods, such as Differential Ghost Imaging (DGI) [30], Normalized Ghost Imaging (NGI) [31], and Second-order Coherence Normalizing Ghost Imaging (SGI) [39], utilize a second bucket detector to record a reference signal which is used in reconstruction to weight the correlation signal. The reference signal contains information from not only illumination patterns but also illumination intensity. The information of illumination intensity recorded by the reference arm is helpful to suppress the impact of illumination fluctuation. These methods have been proven to be equivalent to each other in terms of their effectiveness in suppressing light fluctuations in Shuai Sun’s work [39]. Take NGI for example, the reference signal can be expressed as
(11)$$R_{i}^{*}=I_{i}\overset{N}{\underset{j=1}{\sum }}P_{i,j}$$

The signal used for reconstruction is normalized by the reference signal, namely
(12)$$\frac{S_{i}{}^{*}}{R_{i}^{*}}=\frac{1}{\sum_{j=1}^{N}P_{i,j}}\overset{N}{\underset{j}{\sum }}P_{j}O_{j}$$

Both the signal in the reference arm and the signal in the detection arm will be affected by the fluctuations in illumination power, and thus the contribution to the reconstruction will be weighted more appropriately. Jeffrey H. Shapiro has proven ghost imaging can be performed with only a bucket (single-pixel) detector [11], which means the reference arm is not necessary. The architecture of Computational Ghost Imaging (CGI) is concise and enhances its process of integration and practicability. However, the system without reference arm is less resistant to illumination fluctuation. We propose a normalization protocol that can suppress the impact of illumination fluctuation without a second detector.

Due to the inconsistence, the difference among observed values of different measurements cannot be ignored. However, the interval of time of the two inverse measurements is short enough that their corresponding illumination intensity can be considered the same, namely
(13)$$I_{i+}\approx I_{i-}=I_{i},i=1,2,3,\ldots ,N$$

Considering the way two inverse Hadamard patterns are generated, they have the following properties
(14)$$P_{i+,j}+P_{i-,j}=1$$

Naturally, the two inverse measurements can also be summed as
(15)$$S_{i+}{}^{*}+S_{i-}{}^{*}-2S_{1-}{}^{*}=I_{i}\overset{N}{\underset{j}{\sum }}O_{j}$$

Note that the measured value $S_{1-}{}^{*}$ is corresponding to the pattern $P_{1-}$ whose elements are all ‘0’. Thus, the measured value $S_{1-}{}^{*}$ can be a rough estimate of additive noise. In Equation (15), $\overset{N}{\underset{j}{\sum }}O_{j}$ is a constant that depends on the image to be obtained. Therefore, the sum of two inverse measurement signal values is proportional to the illumination intensity. As shown in Figure 2c, it reflects how the illumination intensity fluctuates. The ratio of the difference and the sum can eliminate the impact of illumination intensity fluctuation, as shown in Figure 2d. From this, the sum of two inverse measurement signal values can be used as the weight to normalize measurement. This normalization protocol can be simply formulated as
(16)$$\frac{S_{i+}{}^{*}-S_{i-}{}^{*}}{S_{i+}{}^{*}+S_{i-}{}^{*}-2S_{1-}{}^{*}}=\frac{1}{\sum_{j}^{N}O_{j}}\overset{N}{\underset{j}{\sum }}P_{j}O_{j}$$

The illumination intensity is a function of time. This relationship can be represented by the frequency and amplitude of the function. Therefore, the improvement brought by the normalization protocol will be affected by the frequency and amplitude. More specifically, the frequency here should be the relative magnitude of light intensity fluctuation frequency to the frequency of SLM. The intensity of an actual light source consists of countless frequency components. Fortunately, the imaging process of HSI is linear transformation, separate analysis of different frequency components is feasible according to Fourier transform.

The initial test is performed through simulation. In this simulation, illumination intensity is a simple sine wave whose amplitude is 0.15 and frequency is set as 100 Hz, and the SLM is operated at 20 kHz. We imaged a modified US Air Force (USAF) resolution test chart and set the imaging resolution to 128 × 128. Three different imaging strategies are used to reconstruct images: (a) Hadamard single-pixel imaging without differential or normalization (HSI). Instead of the original Hadamard matrixes that contain ‘−1’, we use the patterns with a ‘+’ subscript in the patterns used in DHSI, namely $P_{i+}$, which contain only ‘0’ and ‘1’; (b) Differential Hadamard single-pixel imaging (DHSI); (c) Differential Hadamard single-pixel imaging with reference arm (RDHSI); and (d) Normalized-Differential Hadamard single-pixel imaging (NDHSI), namely DHSI with proposed normalization protocol. The quality of the reconstructed image is evaluated by the contrast-to-noise ratio (*CNR*), which is defined by
(17)$$CNR=\frac{\langle I_{f}\rangle -\langle I_{b}\rangle }{\frac{1}{2}\left(\sigma_{f}+\sigma_{b}\right)}$$
where $\langle I_{f}\rangle$ is the average intensity of the feature, $\langle I_{b}\rangle$ is the average intensity of the background, and $\sigma_{f}$ and $\sigma_{b}$ are the standard deviations of the intensities in the feature and the background, respectively. Here, $\langle I_{f}\rangle$ is calculated from the data within the white block, highlighted by a solid red square in Figure 3d, $\langle I_{b}\rangle$ is calculated from the data highlighted by the blue square in Figure 3d.

Reconstructed images and *CNR* of them are shown in Figure 3, and the comparison of the measurements and calculation amount is shown in Table 1. Here the “Number of multiplications” and “Number of additions” in Table 1 represent the amount of computation required to reconstruct a single image by different imaging strategies.

The results in Figure 3 and Table 1 show that this normalization protocol can achieve results of quality close to correction by reference signal. Both RDHSI and NDHSI bring improvement of CNR: the former requires a reference with a second detector, while the latter requires only a little more calculation. Note that all the data used to reconstruct images in NDHSI are the same as DHSI, which means no additional hardware and no additional operation is required.

## 3. Results

To test our method for image reconstruction, an experiment was performed. Figure 4 illustrates our imaging system. We used a 532 nm-single longitudinal mode laser with 200 mw rated power from Beijing Viasho Technology as the light source and we used a digital micromirror device (DMD), which is produced by Texas Instruments, to modulate the light field. It consisted of an array of 1024 × 768 micro-mirrors and operated at 22 kHz. The light signals were detected by the detector PDA100A2. This detector produced by Thorlabs was an amplified, switchable-gain, silicon (Si) detector designed for the detection of light signals ranging from 320 nm to 1100 nm. The signals were recorded and transformed to the computer by the sampling scope which is Pico-Scope 9300 produced by Pico Technology with up to 25 GHz bandwidth.

A beam splitter and a second detector were used in the experiment to record illuminance fluctuations. The recorded signal can be regarded as reference signal in RDHSI. We imaged the resolution test chart used in the simulation and set the imaging resolution to 128 × 128. Thus, 32,768 (128 × 128 × 2) Hadamard patterns were generated in a manner of Russian Dolls ordering [40]. These patterns were preloaded and stored in the 2 GB memory of DMD. DMD operates at 20 KHz, which means each pattern was displayed for 50 μs. Pico-Scope operates at 1 MHz so that 50 sampling points could be obtained during each pattern displayed, and $S_{i}$ is the average of the middle 80% sampling points. To simulate the fluctuating illumination, we used a computer program to control the drive current to change the laser output light intensity. The relation between the drive current and the output power of laser is nonlinear, thus the signal recorded by the second detector was the better choice for characterizing the fluctuation in light intensity. We describe the illumination fluctuation level with
(18)$$\omega =\frac{\sigma_{I}}{\langle I\rangle }\times 100\%$$
where $\langle I\rangle$ and $\sigma_{I}$ are the average and standard deviations of the light intensity versus time respectively. When imaging, the light travels into the DMD through a hole on its side. Then, the modulated light reflects on the object and the total intensity was detected by PDA, eventually sampled by the sampling oscilloscopes. After sampling, we computed the average then reconstruct the image and calculate its *CNR*.

Reconstructed images by various strategies in different illumination fluctuation levels are given in Figure 5a–f. By calculating the *CNR* of the images, the sensitivity to the illumination fluctuation of each strategy is embodied. As shown in Figure 5, the quality of the reconstructed image by HSI worsens as the illumination fluctuation level increases, and other approaches show the same trend. The SNR of the signal detected by the detector decreased when the illumination temporal fluctuation increases, thus more small signals were covered by the noise. Figure 5a shows the images reconstructed when keeping the drive current of the laser constant. All four images are clear with little noise, yet we can determine the noise immunity according to the *CNR* of them. As expected, the latter three approaches perform better than HSI without differential operations. However, normalization seems to bring no further improvement. This comes from the fact that additive noise tends to dominate when the illumination fluctuation stays low. The differential operation was conducive to suppression of additive noise, hence reconstruction with differential brought more significant improvement compared with normalization which deals with multiplicative noise caused by illumination fluctuation. The results of the other two low light level groups shown in Figure 5b,c also bear this out.

Things change as the illumination fluctuation level continues to increase. When the illumination fluctuation level comes to 8.17%, as shown in Figure 5d, visible noise appears in the image reconstructed by HSI whose *CNR* is less than 3. Images obtained by DHSI, RDHSI, and NDHSI maintain clarity; however, their difference becomes more obvious: the *CNR* of the image reconstructed by NDHSI was 10.88, which is 46% higher than the *CNR* of the image reconstructed by DHSI. RDHSI achieved the best image quality, and NDHSI was close to it.

To demonstrate the improvement brought by our protocol, the line profiles of the images in Figure 5g,h when the illumination fluctuation level comes to 8.17% are illustrated in Figure 5i,j. Figure 5i shows image reconstructed with normalization is in better agreement with the original image in bright and dark areas, and Figure 5j shows the same situation in the area that contains both bright and dark. As expected, NDHSI achieves results of quality close to RDHSI.

The line profiles of the images in Figure 5g,h also show how the illumination fluctuations affect the reconstructed images. The fluctuations of the illumination, whose nature is the fluctuations of the light source’s output power, result in different levels of deviation from the true value, thus noise in the reconstructed images.

Normalization operation can still enhance image quality significantly though the differential operation already brings dramatic improvement. This is because the contribution to the noise of such severe light fluctuations cannot be ignored. Naturally, the higher the level of illumination fluctuation is, the higher proportion of total noise the illumination noise accounts for. As shown in Figure 5e,f, the gap between the quality of images retrieved with and without normalization operation became wider when the illumination fluctuation level increased, and NDHSI still achieved results of quality close to RDHSI.

Figure 6 shows the experimental results with an incandescent lamp. According to the illumination intensity recorded by the reference arm, the incandescent bulb flickers at 100 Hz, and the fluctuation level is 6.93%. The sum of differential signals shows a great consistency with the light intensity, as shown in Figure 6c. The average ***RMSE*** between the normalized reference signals and the normalized sums of differential signals is only 2.4%. As a result, the quality of the image reconstructed by NDHSI is close to the RDHSI, which is better than DHSI.

It is worth mentioning that this normalization protocol Is enlightened by single-pixel imaging based on the Hadamard transform, but it can also be applied in other single-pixel imaging technologies with differential operation since no additional hardware and no more measurements are required.

## 4. Conclusions

In conclusion, we present a new normalization protocol based on Hadamard single-pixel imaging, which is helpful to reduce the noise caused by illumination intensity fluctuation. Then, we compare the quality of reconstructed images using methods with and without normalization or differential operations. It turns out our normalization protocol can bring significant improvement, especially at high illumination fluctuation levels. In the experiment of imaging USAF resolution test chart with 128 × 128 resolution, when the fluctuation of the laser output intensity was 27.65%, our normalization protocol brought 112% improvement in CNR. Moreover, our normalization requires only a little extra calculation but no additional hardware and no additional operation. The normalization protocol proposed is compatible with any single-pixel imaging system, thus it has the potential to be feasible across the electromagnetic spectrum, especially when those low-power sources which are more sensitive to temperature and thus suffer more from the fluctuation of illumination are used in nonvisible imaging, such as Globar [17,18] in infrared imaging and spintronic terahertz emitter [23] in terahertz imaging.

## Figures and Tables

**Figure 1 sensors-23-01478-f001:**
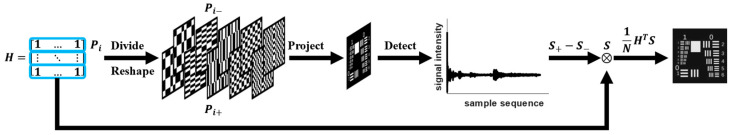
Schematic diagram of DHSI. Each row of Hadamard is reshaped to the size of the image and then divided into two patterns that are comprised of elements ‘1’ and ‘0’. During the measurements, one pattern ($P_{i+}$) is displayed and followed immediately by its inverse ($P_{i-}$). Their corresponding detected signals are recorded and used to produce a differential signal, which is eventually used for image reconstruction.

**Figure 2 sensors-23-01478-f002:**
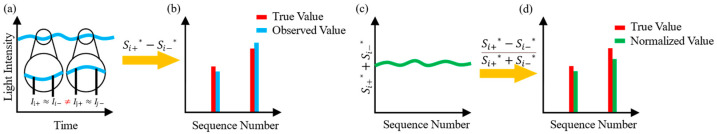
Schematic diagram of normalization protocol. $S_{i+}{}^{*}$ and $S_{i-}{}^{*}$ represent the observed values corresponding to the true values $S_{i+}$ and $S_{i-}$, respectively. (**a**) Illumination intensity fluctuates over time, while two inverse measurements are adjacent so that their corresponding illumination intensities are close to each other; (**b**) observed value (the blue column) is inconsistent with the true value (the red column) in DHSI due to the illumination fluctuation; (**c**) the sum of differential signals shows great consistency with the illumination intensity; (**d**) normalized value (the green column) is proportional to the true value (the red column).

**Figure 3 sensors-23-01478-f003:**
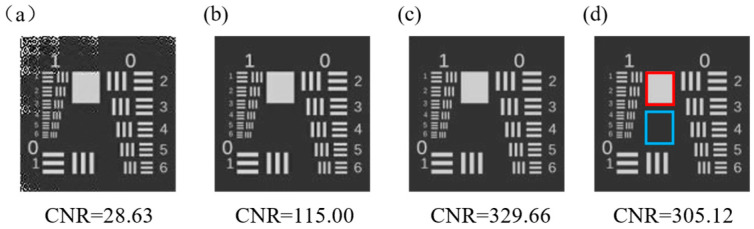
Reconstructed images of USAF resolution test chart utilizing (**a**) HSI, (**b**) DHSI, (**c**) RDHSI, and (**d**) NDHSI when the illumination fluctuation’s amplitude is 0.15 and the frequency is 100 Hz. The imaging resolution to 128 × 128 and thus 16,384 patterns are used in HSI and 32,768 patterns are used in DHSI, RDHSI, and NDHSI.

**Figure 4 sensors-23-01478-f004:**
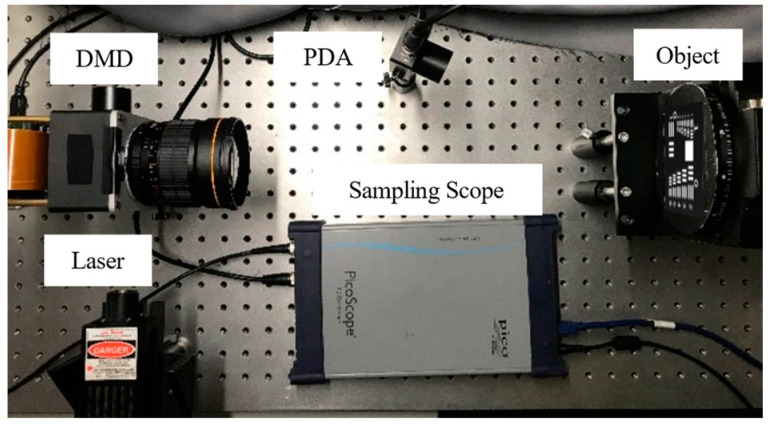
NDHSI system setup. The light emitted from the laser (VA-I-SLM-532, 532 nm, 200 mW) is modulated and reflected on DMD (DLPC410, micromirror array 1024 × 768, pitch 13.7μm, operating at 20 KHz), and then the structured light is projected on the object by a camera lens (Nikon AF Nikkor, f = 50 mm, F = 1.8 D). The total intensity of light reflected by the object is recorded by the Si free-space amplified photodetector (PDA100A2). The signals were recorded and transformed to the computer by the sampling scope (Pico-Scope 9300, 25 GHz). Combining the knowledge of the modulation patterns and the corresponding intensity, images can be reconstructed.

**Figure 5 sensors-23-01478-f005:**
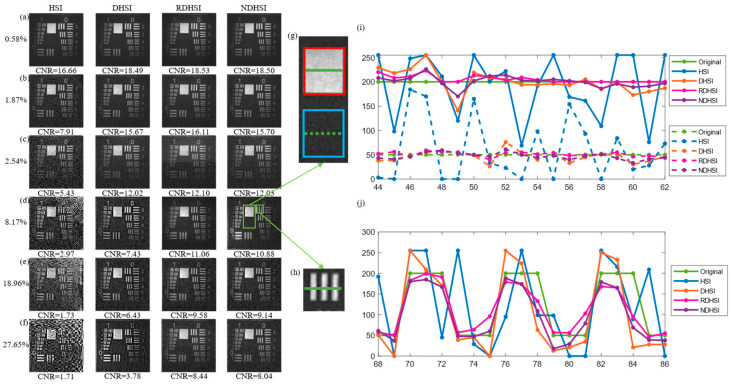
Experimental results with laser. Reconstructed images of USAF resolution test chart utilizing HSI, DHSI, RDHSI, and NDHSI when the illumination fluctuation level is (**a**) 0.6%; (**b**) 1.87%; (**c**) 2.54%; (**d**) 8.17%; (**e**) 18.96%; (**f**) 27.65%; respectively. (**g**) Magnified view of bright and dark areas. (**h**) Magnified view of the area that contains both bright and dark. (**i**) Line profiles of the images highlighted by the solid green line (in the bright area) and the dashed green line (in the dark area) in (**g**) when the illumination fluctuation level is 8.17%. (**j**) Line profiles of the images are highlighted by the solid green line in (**h**) when the illumination fluctuation level is 8.17%.

**Figure 6 sensors-23-01478-f006:**
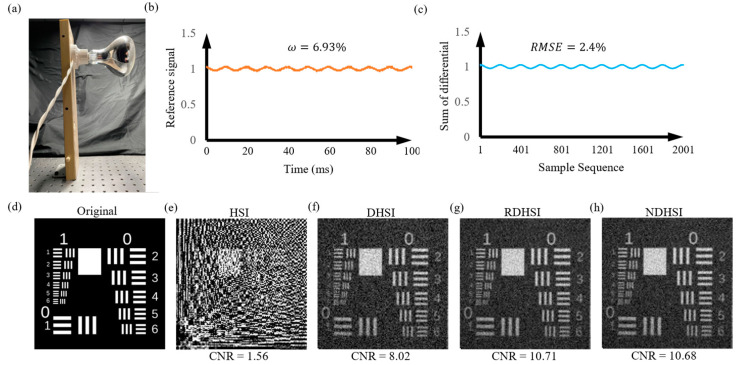
Experimental results with an incandescent lamp. (**a**) The incandescent lamp used in the experiment. (**b**) The illumination intensity is recorded by the reference arm during 0–100 ms, the period of the bulb’s flicker is 10 ms due to the alternating current working at 50 Hz. Illumination fluctuation level is 6.93%. (**c**) Part of the normalized sum of the differential signals, corresponding to 0–100 ms. The average ***RMSE*** between the normalized reference signals and the normalized sums of differential signals is only 2.4%. (**d**) Original USAF resolution test chart. Reconstructed images utilizing (**e**) HSI, (**f**) DHSI, (**g**) RDHSI, and (**h**) NDHSI.

**Table 1 sensors-23-01478-t001:** Comparison of the four different imaging strategies.

**Type**	HSI	DHSI	RDHSI	NDHSI
Number of measurements	$N^{2}$	$N^{2}$	$N^{2}$	$N^{2}$
Number of multiplications	$N^{2}$	$N^{2}$	$N^{2}+N$	$N^{2}+N$
Number of additions	$N^{2}-N$	$N^{2}$	$N^{2}$	$N^{2}+N$
Reference arm	No	No	Yes	No

## Data Availability

No new data were created.

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
