# Peer review of "Illumination Temporal Fluctuation Suppression for Single-Pixel Imaging"

_sensors, 2023, doi:10.3390/s23031478_

Round 1

Reviewer 1 Report

In this manuscript, the authors introduce a normalization protocol based on Differential Hadamard Single-pixel Imaging (DHSI) to suppress the illumination temporal fluctuation. Some theoretical analysis and experimental results are presented, which show that the proposed method can achieve better reconstruction image without additional devices.

There are some very commendable points in this article. First, the structure is reasonable and logical, and the concise protocol is clearly and moderately illustrated. Second, the results are clearly arranged and appears correct. Comparison among different schemes makes the improvement more intuitive, and the experiments utilizing incandescent lamps make it more convincing.

However, there are a number of issues which need to be further addressed. First, the analysis of the results is superficial and not comprehensive enough. Second, the authors should pay attention to English grammar, spelling, and sentence structure so that the goals and results of the study are clear to the reader.

Author Response

Response to Reviewer 1 Comments

Dear reviewer,

Thank you for giving us the opportunity to submit a revised draft of the manuscript for publication in Sensors. We appreciate the time and effort that you and the reviewers dedicated to providing feedback on our manuscript and are grateful for the insightful comments on and valuable improvements to our paper. We have incorporated the suggestions made by the reviewers. Those changes are highlighted in the manuscript by using the track changes mode in MS Word. Please see below, for a point-by-point response to the reviewers’ comments and concerns. All line numbers and page numbers refer to the revised manuscript file with tracked changes. This response to the reviewer’s comments is also uploaded as a word file, please see the attachment.

Point 1: In this manuscript, the authors introduce a normalization protocol based on Differential Hadamard Single-pixel Imaging (DHSI) to suppress the illumination temporal fluctuation. Some theoretical analysis and experimental results are presented, which show that the proposed method can achieve better reconstruction image without additional devices.

There are some very commendable points in this article. First, the structure is reasonable and logical, and the concise protocol is clearly and moderately illustrated. Second, the results are clearly arranged and appears correct. Comparison among different schemes makes the improvement more intuitive, and the experiments utilizing incandescent lamps make it more convincing.

Response 1: Thank you again for your positive comments.

Point 2: However, there are a number of issues which need to be further addressed. First, the analysis of the results is superficial and not comprehensive enough. Second, the authors should pay attention to English grammar, spelling, and sentence structure so that the goals and results of the study are clear to the reader.

Response 2: Thank you for pointing this out. More detailed analysis of experiment has been added in Section 3 Results.

Thank you for pointing this out. We have tried our best to polish the language in the revised manuscript.

Reviewer 2 Report

The manuscript proposed a method in which the Differential Hadamard patterns were normalized in reconstruction process. Therefore, the illumination temporal fluctuations were suppressed and the images’ quality was significantly improved. Before publication, the following issues need to be considered by the authors.

1.      In line 127, the authors mentioned that existing temporal correction methods such as DGI, NGI SGI, TCGI have been proved to be equivalent to each other in terms of their effectiveness in suppressing light fluctuations. Is this point of view supported by relevant literatures?

2.      In line 126, the authors mentioned that the algorithms utilize a second bucket detector to record a reference signal, and in line 131, the authors mentioned that the reference arm is needed for temporal illumination fluctuation correction in methods mentioned above, which is not an accurate statement. The reference signal (illumination patterns) is indispensable in no matter what algorithms or scheme based on the physical principle of ghost imaging and single-pixel imaging while the second detector and reference arm are not. In addition, is the algorithm utilized in this manuscript compressed sensing?

3.       According to functions (7) and (8), the noise n_i may not be ignored in function (15).

4.      The explanation of the influence on the images’ quality caused by illumination temporal fluctuation seems incomplete. Figure 2 and the related paragraph on Page 13 didn’t explain why the imaging quality of RDHSI and NDHSI still decreased when the illumination temporal fluctuation increased.

5.      The meaning of “Number of multiplications” and “Number of additions” are ambiguous in Table 1.

Author Response

Response to Reviewer 2 Comments

Dear reviewer,

Thank you for giving us the opportunity to submit a revised draft of the manuscript for publication in Sensors. We appreciate the time and effort that you and the reviewers dedicated to providing feedback on our manuscript and are grateful for the insightful comments on and valuable improvements to our paper. We have incorporated the suggestions made by the reviewers. Those changes are highlighted in the manuscript by using the track changes mode in MS Word. Please see below, for a point-by-point response to the reviewers’ comments and concerns. All line numbers and page numbers refer to the revised manuscript file with tracked changes. This response to the reviewer’s comments is also uploaded as a word file, please see the attachment.

Point 1: In line 127, the authors mentioned that existing temporal correction methods such as DGI, NGI SGI, TCGI have been proved to be equivalent to each other in terms of their effectiveness in suppressing light fluctuations. Is this point of view supported by relevant literatures?

Response 1: Thank you for this suggestion. We have added corresponding explanation and reference in line 127 to make this clear. This point of view is supported by Ghost imaging normalized by second-order coherence (doi.org/10.1364/OL.44.005993). In that paper, theory calculation and experimental results indicate that “the error caused by the fluctuation of illumination power can be reduced in DGI, and also in NGI and SGI, both of which are equivalent to DGI in this case.” (In the second paragraph from bottom).

Point 2: In line 126, the authors mentioned that the algorithms utilize a second bucket detector to record a reference signal, and in line 131, the authors mentioned that the reference arm is needed for temporal illumination fluctuation correction in methods mentioned above, which is not an accurate statement. The reference signal (illumination patterns) is indispensable in no matter what algorithms or scheme based on the physical principle of ghost imaging and single-pixel imaging while the second detector and reference arm are not. In addition, is the algorithm utilized in this manuscript compressed sensing?

Response 2: Thank you for pointing this out. We have revised the statement in Section 2.3 Normalization protocol. “The reference signal” is replaced with “the information of illumination intensity recorded by the reference arm” to avoid confusion. The reference signal contains the information from not only illumination patterns but also illumination intensity. The information of illumination intensity recorded by the reference arm is helpful to suppress the impact of illumination fluctuation. Compressed sensing is not utilized in the algorithm.

Point 3: According to functions (7) and (8), the noise n_i may not be ignored in function (15).

Response 3: We feel sorry for our carelessness. In our experiment, we used S1-* as a rough estimate of additive noise to eliminate its impact. All the elements of the second pattern, namely P1-, are zeros, thus the measured value S1-* can be a rough estimate of additive noise. The function (15) and (16) in Section 2.3 Normalization protocol have been corrected, and corresponding explanation is also added in line 155.

Point 4: The explanation of the influence on the images’ quality caused by illumination temporal fluctuation seems incomplete. Figure 2 and the related paragraph on Page 13 didn’t explain why the imaging quality of RDHSI and NDHSI still decreased when the illumination temporal fluctuation increased.

Response 4: We sincerely appreciate the valuable comments. Explanation is added in line 232. SNR of the signal detected by the detector decrease when the illumination temporal fluctuation increased, more small signals will be covered by the noise, thus the imaging quality of all methods decreased when the illumination temporal fluctuation increased.

Point 5: The meaning of “Number of multiplications” and “Number of additions” are ambiguous in Table 1.

Response 5: Thank you for pointing this out. We have added explanation of the meaning of “Number of multiplications” and “Number of additions” in line 190.“Number of multiplications” and “Number of additions” in Table 1 represent the amount of computation required to reconstruct a single image by different imaging strategies.

Reviewer 3 Report

Comments attached.

Author Response

Response to Reviewer 3 Comments

Dear reviewer,

Thank you for giving us the opportunity to submit a revised draft of the manuscript for publication in Sensors. We appreciate the time and effort that you and the reviewers dedicated to providing feedback on our manuscript and are grateful for the insightful comments on and valuable improvements to our paper. We have incorporated the suggestions made by the reviewers. Those changes are highlighted in the manuscript by using the track changes mode in MS Word. Please see below, for a point-by-point response to the reviewers’ comments and concerns. All line numbers and page numbers refer to the revised manuscript file with tracked changes. This response to the reviewer’s comments is also uploaded as a word file, please see the attachment.

Point 1: In the present manuscript the authors propose normalization protocol in differential measurements in case of single-pixel imaging. In my opinion the manuscript needs rigorous revision before it can be accepted for publication Following points need to be clearly addressed to bring out its novelty in case of single-pixel imaging using normalization protocol.

I am not convinced by their argument that adopted normalization is better as similar results are obtained using simply reference arm. Further it is not clear how their ‘normalization protocol based on Hadamard single-pixel imaging (NDHSI) is helpful to reduce the noise caused by illumination intensity fluctuation” as they have claimed. Similar results are obtained using RDHSI.

Response 1: Thank you for pointing this out. We have added the advantage of normalization in contrast to the reference arm in section 2.3 Normalization protocol. The normalization can be adopted to the system without reference arm, whose architecture is more concise and easier for integration and practicability.

Thank you for pointing this out. We have added more detailed description of how the normalization protocol is helpful to reduce the noise caused by illumination intensity fluctuation. The sum of two inverse measurement signal values is used as an estimate of illumination intensity to eliminate its impact.

Point 2: In conclusion their argument “Therefore, the proposed normalization protocol can be expected to be more effective in nonvisible imaging. However, this normalization protocol 292is effective only if the illumination intensities corresponding to the two inverse measurements are close enough, hence it is unable to deal with high-frequency fluctuation” is too speculative and has no substantial evidence.

Response 2: Thank you for pointing this out. The speculative argument in conclusion has been removed.

Point 3: A critical comparison with other adopted approaches e.g. discrete cosine transform (DCT) has to be there to bring out the importance of the approach particularly in noisy conditions (Scientific Reports; (2020) 10:19451https://doi.org/10.1038/s41598-020-76487-3). They should critically discuss other approaches besides differential Hadamard single-pixel imaging (DHSI).

Response 3: Thank you for pointing this out. Although we agree that this is an important consideration, it is beyond the scope in this manuscript. Type of approach will affect the quality of the reconstructed image, which is unfavorable for the analysis of the effect of normalization protocol. The difference between different approaches is not the focus of our work.

Point 4: Effect of illumination fluctuations need to be elaborated. It is not clear from the study what the nature of temporal fluctuations is.

Response 4: Thank you for pointing this out. More elaborated analysis of temporal illumination fluctuations and its impact has been added in Section 2.2 Noise in DHSI and Section 3 Results. The fluctuations of the illumination, whose nature is the fluctuations of light source’s output power, result in different levels of deviation from the true value, thus noise in the reconstructed images. The nature of the temporal fluctuations is the light source’s output power fluctuations, which may be caused by the fluctuations of temperature, drive current and other factors.

Point 5: More details of the experiment should be provided.

Response 5: Thank you for this suggestion. We have added more details of the experiment in Section 3 Results.

Point 6: Figure caption should be descriptive which is not there in the reported paper.

Response 6: Thank you for this suggestion. We have added descriptive figure caption for each figure.

Point 7: English also needs to be improved significantly.

Response 7: Thanks for your suggestion. We have tried our best to polish the language in the revised manuscript.

Point 8: Reference 17, authors sequence has to be corrected

Response 8: Thank you for pointing this out. We have corrected the authors sequence in reference 17.

Round 2

Reviewer 3 Report

The authors have addressed the comments and have revised the manuscript significantly. Hence, in my opinion it can be accepted for publication. However, before final publication , In the caption of Figure 6 (L303), 'works' should be replaced by 'working'.